# Genetic Biomarkers of Panic Disorder: A Systematic Review

**DOI:** 10.3390/genes11111310

**Published:** 2020-11-04

**Authors:** Artemii Tretiakov, Alena Malakhova, Elena Naumova, Olga Rudko, Eugene Klimov

**Affiliations:** 1Faculty of Biology, Lomonosov Moscow State University, 119991 Moscow, Russia; temanch@gmail.com (A.T.); alenaafonchikova@gmail.com (A.M.); naumova@mail.bio.msu.ru (E.N.); rudko@mail.bio.msu.ru (O.R.); 2Center of Genetics and Life Sciences, Sirius University of Science and Technology, 354340 Sochi, Russia

**Keywords:** panic disorder, genetic biomarker, anxiety disorders

## Abstract

(1) Background: Although panic disorder (PD) is one of the most common anxiety disorders severely impacting quality of life, no effective genetic testing exists; known data on possible genetic biomarkers is often scattered and unsystematic which complicates further studies. (2) Methods: We used PathwayStudio 12.3 (Elsevier, The Netherlands) to acquire literature data for further manual review and analysis. 229 articles were extracted, 55 articles reporting associations, and 32 articles reporting no associations were finally selected. (3) Results: We provide exhaustive information on genetic biomarkers associated with PD known in the scientific literature. Data is presented in two tables. Genes *COMT* and *SLC6A4* may be considered the most promising for PD diagnostic to date. (4) Conclusions: This review illustrates current progress in association studies of PD and may indicate possible molecular mechanisms of its pathogenesis. This is a possible basis for data analysis, novel experimental studies, or developing test systems and personalized treatment approaches.

## 1. Introduction

Anxiety disorders are a very wide group of mental disorders which includes, according to ICD-11 (World Health Organization, 2018), a variety of conditions with panic disorder (PD) being one of the most common and often chronic [1,2]. It is predominantly characterized by panic attacks and anticipatory anxiety. Many researchers report that PD can noticeably reduce the quality of a patient’s life [3]. Moreover, Sherbourne et al. (1996) report that PD may alter a patient’s quality of life is even more dramatically than other severe chronic diseases including diabetes, cardiovascular and lung diseases [4].

The prevalence of PD is estimated at 1–3%, with females suffering from it twice as frequently as males [5]. Family and twin studies estimated PD heritability at 0.48 [6], with a relative risk of PD in proband’s first-degree relatives being 6–17 times higher than average risk across the whole population [7,8].

Concordance in monozygotic twins is approximately 20.7–73.0% [6]. Both genetic and environmental factors are considered to be involved in the development of PD. It has been shown that being in stressful conditions can provoke panic attacks in people predisposed to anxiety [9].

Molecular genetic markers, along with psychological examination and evaluation of environmental factors causing stress, are promising in regard to creating effective PD-predisposition tests. This is especially important for people who are frequently exposed to stressful conditions due to their profession and/or lifestyle. Development of such predisposition tests may help to take PD-risks into account and preventing the disease thus increasing the quality of life and professional effectiveness in both the general population and trained professionals.

In this work, we aimed to conduct a search and analysis of currently available literature data on PD-associated genetic markers.

## 2. Materials and Methods

We used Pathway Studio 12.3 software (https://mammalcedfx.pathwaystudio.com/, Elsevier, The Netherlands) for automated steps of our analysis. Pathway Studio uses the built-in ResNet database (https://mammalcedfx.pathwaystudio.com/, Elsevier) which is a curated database containing extracted from scientific literature via data-mining information regarding biological objects and their interactions. The database is updated every two weeks, the latest update included in the current study took place on 1st September 2020. According to Elsevier (https://www.elsevier.com/__data/assets/pdf_file/0017/91601/ELS_PathwayStudio-Fact-Sheet_Transform-Your-Research_June-16.pdf), Pathway Studio has access to more than 4.0M of data-mined full-text articles identified by natural language processing algorithms biological relationships.

The search algorithm for PD biomarkers was as follows:Searching for genes (proteins) linked to PD through reported polymorphism associations. The initial list of genes was acquired using PathwayStudio 12.3 software. Search was performed by adding “Panic Disorder” object and extracting all proteins and complexes linked to “Panic Disorder”. Linkage type: GeneticChange was used. This type allows to search for reported polymorphism-condition associations. In accordance to ResNet and Pathway Studio features, the acquired list included not only protein-encoding genes but microRNA-encoding genes as well, if they are reportedly linked to PD.Acquiring a reference table via “View relation table”. This function provides access to reference IDs and data-mined “Sentences” containing association reports from original articles. On this step additional search was performed using PubMed (http://www.ncbi.nlm.nih.gov/pubmed/), TargetInsights (https://demo.elseviertextmining.com/) and GoogleScholar (https://scholar.google.ru/). A total of 229 references supporting each gene-PD link were identified and taken into further screening.Screening was performed by manually analyzing “Sentences” extracted by Pathway Studio from full-text articles. All Sentences were carefully revised by at least three authors, full texts were manually studied for additional details, if necessary. All irrelevant studies identified in this step were excluded from further analysis. A total of 127 irrelevant studies were identified, including 50 literature reviews with no original experiments; 50 articles on irrelevant topics misinterpreted by text-mining algorithms; 27 studies involving model organisms, and lacking human patient studies.Full texts of 102 remaining articles were manually revised and assessed for eligibility by authors. Original research papers with statistically significant results were taken into the further analysis: positive association of certain gene modification (or one of its variants) with the disease; interaction with another gene; association of genotypes or alleles of different genes; the presence of the mutation in families with high occurrence of the disease; cosegregation of genetic alteration with manifestations of the disease.
All articles reporting lack of association were transferred into a separate “No Association” table. This table was analyzed separately (see Step 6). This also included studies reporting an association with the particular symptoms but no association with the disorder in general. A total of 32 articles were excluded due to a lack of significant association.All remaining articles reporting results irrelevant to our study (i.e., treatment response studies) were excluded. A total of 15 articles were excluded due to these reasons.The remaining 55 articles formed the basis of our review. Each paper was manually studied for association availability (significance was assessed by revising *p*-values). Official gene names, polymorphism IDs, sample sizes, *p*-values for each association, and other relevant data were manually extracted and revised. All articles reporting significant associations were considered eligible. Small sample size was not considered an exclusion criterion. However, we are aware that the sample size may result in significant bias. To allow estimating this bias we provide sample sizes in the Sample column.After analyzing the articles reporting associations, we performed a manual analysis of the “No Association” table. The analysis was performed by additional full-text revision and extraction of relevant data.

The algorithm scheme, along with the number of articles added or excluded on each step is presented in Figure 1.

## 3. Results

A total of 229 studies were analyzed, with 127 being excluded during screening and 15 being excluded during other steps. The remaining articles were taken into further analysis and separated into two tables. Table 1 provides the results of the analysis of 55 articles reporting association. Studies reporting lack of association were analyzed separately, results of analyzing 32 studies are provided in Table 2.

## 4. Discussion

Our analysis revealed 40 genes linked to PD. Among them, the majority of associations (in 38 genes) were marked by at least one SNP, 3 (*5HTR2A*, *CCKBR*, *RGS2*) had additional repeat/deletion polymorphism markers, 1 (*COMT*) had additional microsatellite polymorphism marker, and 1 (*GAD1*) had additional methylation alterations associated with PD. The remaining 2 genes (*FOXP3*, *MAOA*) were associated with PD via methylation alterations with no reported associated SNPs.

For the *COMT* gene, its’ link to PD was supported by 9 studies which may be considered the most reliable or most well-studied gene associated with PD. For the majority (25 out of 40) of associations, the link to PD was supported by just 1 study. Associations of polymorphisms in *CCKBR*, *HTR1A*, and *TPH2* genes were supported by 3 studies. Remaining associations in genes *5HTR2A*, *ACE*, *ADORA2A*, *ASIC1*, *CRHR1*, *GAD1*, *MAOA*, *MIR22*, *NPS*, *PDE4B*, and *SLC6A4* were supported by 2 studies.

*COMT* is also among the top genes by the number of associated polymorphisms, however, it only has 5 associated markers, while *SLC6A4* has 8 associated markers and *ADORA2A*, and *TMEM132D* have 6 each.

The majority of genes (16 out of 40) have 2 associated polymorphisms, 7 out of 40 have 3 or 4 associated polymorphisms.

Most of the associated genes (68%) are related to the systems of neurotransmitter synthesis/interaction/degradation: *ACE*, *ADORA2A*, *AVPR1B*, *CCK*, *CCKAR*, *CCKBR*, *CNR1*, *CNR2*, *COMT*, *CRHR1*, *DBI*, *DRD1*, *GABRA5*, *GABRA6*, *GABRB3*, *GAD1*, *GHRL*, *HCRTR2*, *HTR1A*, *HTR2A*, *MAOA*, *NPS*, *NPSR1*, *PBR*, *PGR*, *SLC6A4*, *TPH2*.

The rest of the genes can be attributed to the following functional categories:13%—involved in the regulation of the work of genes (including miRNA); *FOXP3*, *IKBKE*, *MIR22*, *MIR339*, *MIR491*.10%—involved in the reception of extracellular signals and system of secondary messengers functioning; *NTRK3*, *PDE4B*, *RGS2*, *TMEM132D*.5%—involved in the general functioning of neurons; *ASIC1*, *BDNF*.5%—not fitting to any of these categories others; *GLO1*, *MBL2*.

Among the 30 non-associated genes, 16 are present in both tables while 14 are not present in Table 1.

This data represents studies available via ResNet, Google Scholar, PubMed, and TargetInsight databases. Despite the large number of articles covered, several limitations are present. First of all, some articles may be absent from the listed databases. This means that our analysis is not completely exhaustive. Second, only articles available in English were included.

In our work, we summarized literature data on PD biomarkers. Our review may provide a strong base for creating effective test-systems and predisposition tests for early PD diagnostics. This is especially important for people working or living in stressful conditions, as PD is known to be triggered by environmental factors and causes a severe reduction in quality of life [4,93]. Moreover, our study as well as the following studies may be used to reveal possible pathogenesis mechanisms and signal pathways underlying PD. This knowledge is not just theoretical as it may allow developing a personalized approach for each PD case. Personalized medicine is considered a promising field in the case of PD due to the lack of serious advances in treatment in recent years [94]. We believe that recently proposed for personalized psychiatry machine learning approaches already applied to some PD cases [95] may be even more effective if proper genetic screening is conducted. In light of this, reviewing and systemizing PD biomarkers is the way to bring technology and medicine even closer together and revolutionize the field of personalized psychiatry.

## 5. Conclusions

The information summarized in our work illustrates current progress in association studies of panic disorder and may indicate possible molecular mechanisms of the pathogenesis of panic disorder. This information can serve as a basis for both further data analysis, novel experimental studies, or creating effective test-systems and personalizing medicine in the future.

## Figures and Tables

**Figure 1 genes-11-01310-f001:**
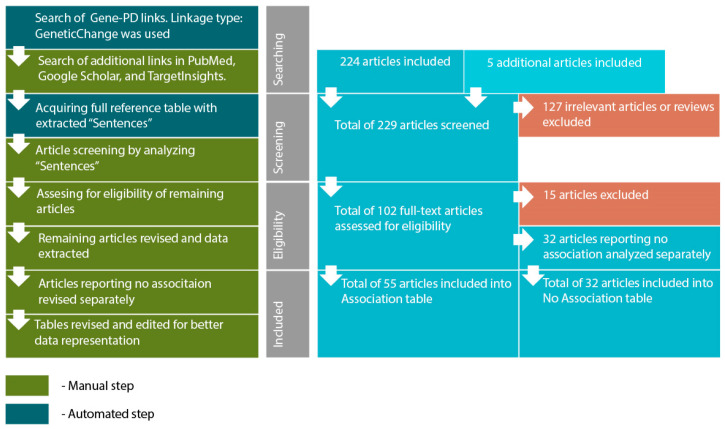
Workflow used in the current review and number of articles included or excluded on each step.

**Table 1 genes-11-01310-t001:** Known panic disorder biomarkers and obtained by text-mining instruments supporting literature data. nP—number of patients, nC—control sample volume.

Gene ^1^	Possible PD Link ^2^	Marker ^3^	Sample ^4^	Commentary ^5^	Reference ^6^
*ACE* (angiotensin I converting enzyme)	ACE peptidase degrades a number of neuropeptides possibly involved in the development of anxious behavior.	rs4646994	nP = 102 (59 female, 43 male), nC = 102 (59 female, 43 male). Germany.	Less active Ins (*p* = 0.0474) allele and genotypes containing Ins (*p* = 0.0195) are associated with PD in male subsample.	[10]
		rs4646994	nP = 43 (28 female and 15 male), nC = 41 (17 female and 24 male). Turkey.	Allele Ins is more frequent in patients (*p* = 0.002). Genotype ID is associated with PD (*p* = 0.003). Patients with allele D had higher risk of respiratory type PD (*p* = 0.034).	[11]
*ADORA2A* (adenosine A2a receptor)	Transmembrane adenosine receptor. Participates in the vasodilation and increasing of neurotransmitter release, which occurs during panic attacks.	rs2298383 and rs685012 (*ASIC1*)	nP = 71 (20 male, 51 female), nC = 100 (37 male, 63 female). Sicily, Italy.	Haplotype rs685012:TT (*ASIC1*) + rs2298383:CT (*ADORA2A*) is more frequent in controls (*p* = 0.026). See also *ASIC1*.	[12]
		rs5751876, rs5751862, rs2298383, rs3761422, rs5760405, rs2267076	nP = 457 (167 male, 290 female), nC = 457 (167 male, 290 female). Germany.	Genotype rs5751876:TT is associated with PD and PD with agoraphobia (*p* = 0.032, *p* = 0.012 respectively). Haplotype GCGCCTTT (rs2298383:C, rs3761422:T, rs5751876:T) is found to be risk (*p* = 0.005), carriers of this haplotype showed higher anxiety levels. Haplotype ATGCTCCC (rs5751862:A, rs2298383:T, rs3761422:C, rs5751876:C), is found to be protective (*p* = 0.006). Allele carriers of rs5751862:G, rs2298383:C, rs3761422:T showed higher levels of anxiety than rs5751862:AA, rs2298383:TT and rs3761422:CC carriers (*p* = 0.001, *p* = 0.002, *p* = 0.002, respectively). A total of 8 SNPs studied.	[13]
*ASIC1* (acid-sensing ion channel subunit 1)	ASIC1 protonsensitive channel is expressed throughout the nervous system and is involved in the formation of fear, anxiety, pain, depression, memory, and learning processes.	rs685012, rs10875995	nP = 414 (232 female, 182 male), nC = 846 (474 female, 372 male). European-American, European-Brazilian.	Alleles rs685012:C and rs10875995:C are more frequent in patients (*p* = 0.0093, *p* = 0.047 respectively). The association is enhanced by a separate analysis of respiratory subtype PD subsample and early development (less than 20 years of age) PD subsample (*p* = 0.027). Neither rs685012 nor rs10875995 was associated with PD in non-respiratory-subtype subsample (*n* = 87) (both *p* > 0.70).	[14]
		rs685012 and rs2298383 (*ADORA2A*)	nP = 71 (20 male, 51 female), nC = 100 (37 male, 63 female). Sicily, Italy.	Allele rs685012:C is more frequent in patients (*p* = 0.030). Haplotype rs685012:TT (*ASIC1*) + rs2298383:CT (*ADORA2A*) is more frequent in controls (*p* = 0.026). See also *ADORA2A*.	[12]
*AVPR1B* (arginine vasopressin receptor 1B)	Arginine receptor which is found in the anterior pituitary and brain structures. Plays a decisive role in modulating the hypothalamic–pituitary–adrenal axis and homeostasis during stress.	rs28632197 and rs878886 (*CRHR1*)	(1) nP = 186 (128 female, 58 male), nC = 299 (217 female, 82 male). Germany. (2) nP = 173 (68 male, 105 female), nC = 495 (195 male, 300 female). Germany.	Genotype-allele haplotype rs28632197:TT (*AVPR1B*) + rs878886:G (*CRHR1*) shows strongest association with PD (*p* = 0.00057). 15 more associations below the significance threshold found.	[15]
*BDNF* (brain-derived neurotrophic factor)	BDNF is one of the most abundant neurotrophins found in brain tissues. It regulates neuronal plasticity: the formation and maintenance of synaptic contacts. May be involved in the pathogenesis of mental diseases including major depression and PD.	rs6265, rs16917204	nP = 136, nC = 263. South Korea.	Haplotype GC (rs6265:G, rs16917204:C) is more frequent in patients (*p* = 0.0009). No association with single SNPs. A total of 3 SNPs studied.	[16]
*CCK* (cholecystokinin)	Cholecystokinins are neuropeptides that play role in satiety and anxiety. CCK-4 tetrapeptide′s ability to induce symptoms of panic attacks is known.	−345g > C	nP = 73 (40 males, 33 females), nC = 252 (133 males, 120 females), Tokyo, Japan.	Rare −345G > C SNP is associated with PD (*p* = 0.025). Other studied SNPs (−36c > t and −188a > g) showed no association.	[17]
		−45C > T, 1270C > G and rs6313, 1438A-G (*5HTR2A*), −94G-A, −800T-C, −48G-A (*DRD1*)	nP = 127 (23 males, 104 females), nC = 146 (37 males, 109 females). Estonia.	Haplotype TG (−45C-T, 1270C-G) is more frequent in patients (*p* = 0.04). No associations with single SNPs. A total of 90 SNPs studied. See also *5HTR2A* and *DRD1*.	[18]
*CCKAR* (cholecystokinin A receptor)	Cholecystokinin A receptor may be involved in the development of PD. Its agonists are known to produce most of the classic symptoms of a panic attack in PD patients.	rs1799723, rs1800908	nP = 109 (64 males, 45 females), nC = 400 (234 males, 166 females). Japan.	Haplotype GT (rs1799723:G, rs1800908:T (−81G/−128T)) is more frequent in patients (*p* < 0.0001). No associations with single SNPs.	[19]
*CCKBR* (cholecystokinin B receptor)	One of the cholecystokinin receptors. Its agonists are known to produce most of the classic symptoms of a panic attack in PD patients.	CT repeat	nP = 111 (71 females, 40 males), nC = 111 (71 females, 40 males). Germany	Diallelic (short (146–162 bp), long (164–180 bp)) analysis: long allele is associated with PD in complete sample (*p* = 0.001); PD with agoraphobia (*p* = 0.002).	[20]
		G > A in the 5′area of the 3′ untranslated region	nP = 99 (63 females, 36 males), nC = 99 (63 females, 36 males). Toronto.	SNP is associated with PD (*p* = 0.004). An analysis using all alleles was also significant (*p* = 0.038). A total of 16 alleles of *CCKBR* and several *CCK* and *CCKAR* polymorphisms were studied.	[21]
*CNR1* (cannabinoid receptor 1)	Endocannabinoid system may play role in the regulation of stress, anxiety, depression, and addictive disorders.	rs12720071, rs806368 and rs2501431, rs2501432 (*CNR2*)	nP = 164 (71% female), nC = 320 (71% female). Spain.	Allele rs12720071:G is associated with PD (*p* = 0.012) and more frequent in females. Total of 12 SNPs studied. See also *CNR2*.	[22]
*CNR2* (cannabinoid receptor 2)		rs2501431, rs2501432 and rs12720071, rs806368 (*CNR1*)	nP = 164 (71% female), nC = 320 (71% female). Spain.	Allele rs2501431:G showed protective effect in male carriers of haplotype A592G (rs2501431) + C315T (rs2501432)) (*p* = 0.043). A total of 12 SNPs were studied. See also *CNR1*.	[22]
*COMT* (catechol-O-methyltransferase)	COMT enzyme is involved in the inactivation of the catecholamine neurotransmitters. The link between COMT activity and depressive and anxiety disorders is known, as well as its involvement in the associated biochemical processes.	D22S944	nP = 693 (613 PD patients from 70 multiplex families, 83 child-parent triads). Caucasian.	Microsatellite D22S944 shows strongest association with PD (*p* = 0.0001–0.0003). Association is present in female subsample (*p* = 0.003), male subsample lacks association. Haplotype rs4680:G, rs4633:C, D22S944 (various variants) is associated with PD (*p* = 0.0001).	[23]
		rs4680 and rs3219151 (*GABRA6*), rs10847832 (*TMEM132D*)	nP = 189 (105 males, 84 females)), nC = 398 (208 males, 190 females). Japan.	Allele rs4680:G and genotype rs4680:GG are associated with PD (*p* = 5.16 × 10^−4^ and *p* = 8.23 × 10^−6^ respectively). The male subsample lacks this association. A total of 5 SNPs were studied. See also *GABRA6* and *TMEM132D*.	[24]
		rs4680	nP = 178 (91 male, 87 female), nC = 182 (90 male, 92 female). South Korea.	Genotype rs4680:AA increases risk of PD if compared to sum of other genotypes (*p* = 0.042).	[25]
		rs4680	nP = 51 (26 men, 25 women), nC = 45 (23 men, 22 women). South Korea.	Genotype rs4680:AA is associated with PD and increases anxiety (*p* = 0.01).	[26]
		rs4680	(1) 121 nuclear family. Canada. (2) nP = 89 (59 females, 30 males), nC = 89. Canada.	Allele rs4680:G is associated with PD in both samples (*p* = 0.005). In sample 2 association is stronger in female subsample (*p* = 0.008) while males show no association. In subsample of patients with PD with agoraphobia (nP = 68) allele rs4680:G is associated with disease (*p* = 0.01). TDT-test (transmission disequilibrium test) of the 1 sample showed an association of rs4680:G with PD as well (*p* = 0.005). No associations with rs165599 or rs737865.	[27]
		rs4680	nP = 2242. Germany.	Emotional abuse during childhood contributes to the development of PR in men with rs4680:GG genotypes and in women with rs4680: GG and rs4680: GA genotypes (*p* < 0.001 for all cases).	[28]
		rs4680, rs165599	nP = 589 (239 female, 350 male), nC = 539 (196 female, 343 male). USA.	Haplotype rs4680:G + rs165599:A is less frequent in patients of female subsample (*p* = 1.97 × 10^−5^). A total of 10 SNPs studied.	[29]
		rs4680	nP = 105 (66 female, 39 male), nC = 130 (89 female, 41 male). Turkey.	Allele rs4680:A is more frequent in patients (*p* = 0.017). SNP rs6313 (*5HTR2A*) was also studied, no association found.	[30]
		rs4680	nP = 115 (41 males, 74 females), nC = 115 (41 males, 74 females), Germany.	Active allele (472G = V158) is associated with PD (*p* = 0.04), especially in female subsample (*p* = 0.01). Male subsample lacks association (*p* = 1). Interaction with *MAOA* VNTR polymorphism was also studied, no association found.	[31]
*CRHR1* (corticotropin releasing hormone receptor 1)	Corticotropin releasing hormone receptors may be found in the limbic system and in the anterior pituitary gland, which are responsible for human behavior. It is may activate stress- and anxiety-related hormone release.	rs878886 and rs28632197 (*AVPR1B*)	(1) nP = 186 (128 female, 58 male), nC = 299 (217 female, 82 male). Germany. (2) nP = 173 (68 male, 105 female), nC = 495 (195 male, 300 female). Germany.	Genotype-allele pair rs28632197:TT (*AVPR1B*) + rs878886:G (*CRHR1*) shows strongest association with PD (*p* = 0.00057). 15 more associations below the significance threshold found. See also *AVPR1B*.	[15]
		rs17689918	nC = 239 (143 female), nP = 239 (143 female). Germany. nC = 292 (216 female), nP = 292 (216 female). Germany.	SNP rs17689918 is the only significant association with PD which survived all tests in the female subsample (*p* = 0.022). A total of 9 SNPs studied.	[32]
*DBI* (diazepam binding inhibitor)	DBI protein may be found in neuronal and glial cells of the central nervous system. It may be involved in PD via neuroactive steroid synthesis and modulation of GABA channel gating.	rs8192506	nP = 126 (38 male, 88 female), nC = 229 (63 male, 166 female). Germany.	Rare allele rs8192506:G is found to be protective and almost three times more frequent in controls (*p* = 0.032).	[33]
*DRD1* (dopamine D1 receptor)	*DRD1* is known to be linked with depression and anxiety.	−800T-C, −94G-A, −48G-A and 45C > T, 1270C > G (*CCK*), rs6313, 1438A-G (*5HTR2A*)	nP = 127 (23 males, 104 females), nC = 146 (37 males, 109 females). Estonia.	SNP r −94G-A is associated with PD (*p* = 0.02). Haplotype CAA (−800T-C, −94G-A, −48G-A) is found to be protective (*p* = 0.03). A total of 90 SNPs were studied. See also *CCK* and *5HTR2A*.	[18]
*FOXP3* (forkhead- box protein P3 gene)	*FOXP3* hypermethylation may potentially reflect impaired thymus and immunosuppressive Treg function.	Hypermethylation of the *FOXP3* promoter region	nP = 131 (female = 85, male = 44), nC = 131 (female = 85, male = 44). Germany.	Epigenetic study. Lower hypermethylation of the FOXP3 promoter region is found in female PD patient subsample compared to control (*p* = 0.005). The male subsample lacks this association.	[34]
*GABRA6* (γ-aminobutyric acid receptor, α 6 subunit)	GABRA6 subunit is well known for its link to anxiety in mammals. It is the main inhibitor of neurotransmitters in the CNS which regulates several physiological and psychological processes such as anxiety and depression.	rs3219151 and rs4680 (*COMT*), rs10847832 (*TMEM132D*)	nP = 189 (105 males, 84 females)), nC = 398 (208 males, 190 females). Japan.	Allele rs3219151:T and genotype rs3219151:TT are associated with PD (*p* = 2.00 × 10^−7^, *p* = 2.18 × 10^−6^, respectively). Patients carrying genotype rs11060369:AA (*TMEM132D*), and allele rs3219151:T (*GABRA6*) had significantly stronger response to frightening image demonstration (both *p* < 0.01). A total of 5 SNPs were studied. See also *COMT* and *TMEM132D*.	[24]
*GABRA5* (γ-aminobutyric acid type A receptor alpha5 subunit)	The GABA-ergic system is inhibitory. GABA receptors are targets for benzodiazepines that are used for PD treatment.	rs35399885 and rs8024564, rs8025575 (*GABRB3*)	*n* = 1591 (992 samples from 120 multiplex families). Europe.	SNP rs35399885 is associated with PD (*p* = 0.05). A total of 10 SNPs in *GABRA3* and *GABRA5* were studied. See also *GABRA3*.	[35]
*GABRB3* (γ-aminobutyric acid type A receptor alpha3 subunit)		rs8024564, rs8025575 and rs35399885 (*GABRA5*)	*n* = 1591 (992 samples from 120 multiplex families). Europe.	SNPs rs8024564 and rs8025575 are associated with PD (*p* = 0.005, *p* = 0.02 respectively). A total of 10 SNPs in *GABRB3* and *GABRA5* studied. See also *GABRA5*.	[35]
*GAD1* (glutamate decarboxylase 1)	Glutamate decarboxylase 1 is a key enzyme for the synthesis of the inhibitory and anxiolytic neurotransmitter GABA. Suspected of affecting mood and various mental disorders, including anxiety disorders and PD.	CpG hypermethylation	nP = 65 (female = 44, male = 21), nC = 65 (female = 44, male = 21). Germany.	Epigenetic study. *GAD1* methylation levels are lower in patients (*p* = 0.001). Negative life events correlated with a decreased mean methylation in patients—the correlation was observed mainly in the female subsample (*p* = 0.01). A total of 38 methylation sites in promotor/2 intron of *GAD1* and 10 promoter sites in *GAD2* were studied.	[36]
		rs3749034	nP = 478 (female 286, male 192), nC = 584 (female 432, male 152). Germany.	Only rs3749034:A × gender remained significant in a combined sample (*p* = 0.045), but not in the replication sample. A total of 13 SNPs studied.	[37]
*GHRL* (ghrelin and obestatin prepropeptide)	Ghrelin is well-known for its anxiogenic and anxiolytic effects in rodents. Obestatin in contrast, decreases anxiety-like behavior in rodents.	rs4684677	nP = 215 (63 male, 152 female), nC = 451 (199 male, 252 female). Sweden.	Allele rs4684677:A is associated with PD (*p* = 0.025). Two other studied SNPs showed no association.	[38]
*GLO1* (glyoxalase I)	*GLO1* gene expression is known to correlate with anxiety-like behavior in rodents.	rs2736654	nP = 162 (64 male, 98 female), nC = 288 (119 male, 169 female). Italy.	Allele rs2736654:A (Glu) is associated with PD in a subsample of patients without agoraphobia (*p* < 0.025). No associations in the combined sample.	[39]
*HCRTR2* (hypocretin receptor 2)	Hypocretin receptor 2 is expressed exclusively in the brain. Its agonist orexin-A may have an anxiogenic effect in rodents. It is also may be linked to PD via its role in respiration which seems to be altered in PD.	rs2653349	nP = 215 (74 male, 141 female), nC = 454. Sweden.	Allele rs2653349:A is associated with PD in total sample and female subsample (*p* = 0.015, *p* = 0.0015 respectively). Other studied SNP rs2271933 (*HCRTR1*) showed no association.	[40]
*HTR1A* (5-hydroxytryptamine receptor 1A)	*HTR1A* is known to be associated with anxiety disorder and anxiogenic stimuli response.	rs4521432, rs6449693, rs6295, rs13361335	nP = 107 (82 female, 25 male), nC = 125 (88 female, 37 male). Brazil.	Haplotype rs4521432:T, rs6449693:G, rs6295:G, rs13361335:T is associated with PD (*p* = 0.032). Only *HTR1A* showed association with PD (*p* = 0.027). 2 other genes (*SLC6A4* and *HTR2A*) were also studied, no association found.	[41]
		rs6295	nP = 119 (43 male, 73 female), nC = 119 (43 male, 73 female). Japan.	Allele rs6295:G is associated with PD with agoraphobia (*p* = 0.047).	[42]
		rs6295	nP = 133 (49 males, 84 females), nC = 134 (49 males, 85 females). Germany.	Allele rs6295:G is associated with PD with agoraphobia (*p* = 0.03).	[27]
*HTR2A* (5-hydroxytryptamine receptor 2A)	*HTR2A* over- and under-expression is known in PD patients and thus may lead to PD.	102T > C	nP = 63 (29 males, 34 females), nC = 100 (47 males, 53 females). Japan.	SNP 102T > C is associated with PD (*p* = 0.048). Among subsamples, only patients with agoraphobia showed association (nP = 33, *p* = 0.016). Polymorphisms in 3 receptor genes (*HTR1A* (294G > A), *HTR2A* (102T > C), *HTR2C* (23Cys > Ser)) were studied.	[43]
		rs6313 (102T-C), −1438A-G and −45C > T, 1270C > G (*CCK*),−800T-C, −94G-A, −48G-A (*DRD1*)	nP = 127 (23 males, 104 females), nC = 146 (37 males, 109 females). Estonia.	*5HTR2A* polymorphism rs6313:C (102T-C) is associated with pure PD (*p* = 0.01). Haplotype AT (–1438A-G, rs6313:C (102T-C)) is protective (*p* = 0.04). Total of 90 SNPs studied. See also *CCK* and *DRD1*.	[18]
*IKBKE* (inhibitor of nuclear factor kappa B kinase subunit epsilon)	NF-kB protein complex regulates immune system-related genes. The immune system may be involved in the development of anxiety disorders.	rs1539243	nP = 210, nC = 356. Estonia.	Allele rs1539243:T is associated with PD (*p* < 0.001). SNP rs1554286 (*IL10*) was also studied.	[44]
		rs1953090, rs2297543	nP = 190 (44 male, 146 female), nC = 371 (111 male, 260 female). Estonia.	SNPs rs1953090 and rs2297543 are associated with PD in male subsample (*p* = 0.0013, *p* = 0.0456 respectively). A total of 14 SNPs were studied.	[45]
*MAOA* (monoamine oxidase A)	*MAOA* is known for its role in behavior and mental disorders. It plays important role in the metabolism of neuroactive and vasoactive amines in the central nervous system and peripheral tissues.	CpG hypomethylation	nP = 65 (44 females; 21 males), nC = 65 (44 females, 21 males). Germany.	Epigenetic study. Hypomethylation in patients in the female subsample compared to controls (*p* ≤ 0.001). No differences in the male subsample, methylation at 39 out of 42 sites are generally weak or absent. A total of 42 methylation sites were studied.	[46]
		CpG Hypomethylation	nP = 28, females, PD with agoraphobia, nC = 28, females. Europe. nP = 20, females, PD with agoraphobia, nC = 20, females. Europe.	Epigenetic study. Hypomethylation in patients compared to controls (*p* < 0.001). The severity of PR and the degree of methylation are inversely related in patients (*p* = 0.01). A total of 13 methylation sites were studied.	[47]
*MBL2* (mannose-binding lectin 2)	MBL deficiency is the most common hereditary defect in the human innate immune system. It may increase the susceptibility for autoimmune states. PD may be associated with an inflammatory and autoimmune processes.	rs7096206 rs5030737	nP = 1100 (Bipolar disorder = 1000, PD = 100), nC = 349.	Two-marker *MBL2* YA-haplotype (rs7096206, rs5030737) is associated with PD (*p* = 0.0074). No single SNPs were associated with PD. A total of 7 SNPs in genes *MBL2, MASP1, MASP2* were studied.	[48]
*MIR22* (microRNA 22)	MicroRNAs are known to play role in functional differentiation of neurons and their interactions with possible gene-candidates for PD (including *GABRA6*, *CCKBR*, *OMC*, *BDNF*, *HTR2C*, *MAOA*, and *RGS2*).	rs6502892 and rs11763020 (*MIR339*)	(1) nP = 203 (151 women, 52 men), nC = 341 (140 women, 201 men). Spain. (2) nP = 321 (202 women, 119 men), nC = 642 (415 women, 227 men). Finland. (3) nP = 102 (87 women, 25 men), nC = 829 (391 women, 438 men). Estonia.	In Spanish sample: SNP (rs6502892 (*p* < 0.0002) is associated with PD. Associations in other samples lack significance. A total of 712 SNPs in *MIR* genes were studied. See also *MIR339*.	[49]
		rs8076112, rs6502892 and rs4977831, rs2039391 (*MIR491*)	nP = 341 (183 female, 158 male), nC = 229 (128 female, 101 male). South Korea.	SNP rs8076112 is associated with PD (*p* = 0.013). Haplotype rs8076112:C, rs6502892:C is more frequent in patients (*p* = 0.019). ASI-R score is associated with rs6502892 in patients with agoraphobia (*p* = 0.05). See also *MIR491*.	[50]
*MIR339* (microRNA 339)		rs11763020 and rs6502892 (*MIR22*)	(1) nP = 203 (151 women, 52 men), nC = 341 (140 women, 201 men). Spain. (2) nP = 321 (202 women, 119 men), nC = 642 (415 women, 227 men). Finland. (3) nP = 102 (87 women, 25 men), nC = 829 (391 women, 438 men). Estonia.	In Spanish sample: SNPs rs11763020 (*p* < 0.00008) is associated with PD. Associations in other samples lack significance. A total of 712 SNPs in *MIR* genes were studied. See also *MIR22*.	[49]
*MIR491* (microRNA 491)		rs4977831, rs2039391 and rs8076112, rs6502892 (*MIR22*)	nP = 341 (183 female, 158 male), nC = 229 (128 female, 101 male). South Korea.	SNPs rs4977831 (*p* = 0.008) and rs2039391 (*p* = 0.015) are associated with PD. Haplotypes: rs4977831:G, rs2039391:G (*p* = 0.014); rs4977831:A, rs2039391:A (*p* = 0.0002) are more frequent in patients. See also *MIR22*.	[50]
*NPS* (neuropeptide S)	Neuropeptide S can produce behavioral arousal and anxiolytic-like effects in rodents.	rs990310 rs11018195	(1) nP = 183, nC = 315. Spain. (2) nP = 316, nC = 1317. Finland.	SNPs rs990310 and rs11018195, are in linkage disequilibrium and are associated with PD with agoraphobia in Spanish and Finland samples (*p* = 0.021, *p* = 0.022 respectively for Spain; *p* = 0.082, *p* = 0.083 respectively for Finland. A total of 35 SNP in *NPS* and *NPSR1* were studied.	[51]
*NPSR1* (neuropeptide S receptor 1)	*NPSR1* may be linked to signs associated with anxiety and is potentially associated with anxiety disorders via the formation of limbic activity associated with fear.	rs324981	nP = 140 (89 females, 51 males), nC= 245. Japan.	Allele rs324981:T is associated with PD in male subsample (*p* = 0.09).	[52]
*NTRK3* (Tropomyosin receptor kinase C)	*NTRK3* expression change may alter synaptic plasticity leading to abnormal release rates of certain neurotransmitters and thus to altered arousal threshold.	PromII	nP = 59, nC = 86, Spain.	Allele of PromII in 5′UTR-region of *NTRK3* is associated with PD (*p* = 0.02). Patients show a tendency to heterozygosity. 3 other SNPs (IN3, EX5, EX12) were studied, no association found.	[53]
*PBR* (peripheral benzodiazepine receptor)	*PBR* is closely associated with personality traits for anxiety tolerance.	485G > A	nP = 91 (48 Males, 43 females), nC = 178 (90 Males, 88 females). Japan.	Allele G of SNP 485G > A frequencies differ in patients and controls (*p* = 0.014).	[54]
*PDE4B* (phosphodiesterase 4B)	*PDE4B*’s altered expression leads to the change in intracellular cAMP concentrations, which is known in several mental disorders. PDE4B is involved in dopamine-associated and stress-related behaviors.	rs10454453, rs6588190, rs502958, rs1040716	nP = 231 (85 males, 146 females), nC = 407 (162 males, 245 females). Japan.	Allele rs10454453:C is associated with PD in female subsample (*p* = 0.042). Haplotype rs10454453:C, rs6588190:T, rs502958:T, rs1040716:A is associated with PD in complete sample and female subsample (*p* = 0.031). A total of 15 SNPs were studied.	[55]
		rs1040716, rs502958, rs10454453	nP = 94 (75 female and 19 male), nC = (192, 108 female, 84 male). Russia.	Haplotypes: rs1040716:A, T + rs10454453:A + rs502958:A and rs1040716:A, T + rs502958:A are found to be protective (*p* < 0.05).	[56]
*PGR* (progesterone receptor)	Progesterone receptors are present in most brain regions involved in the pathophysiology of panic disorder: brain stem, hypothalamus, hippocampus, amygdala, and raphe nuclei.	rs10895068, ALU insertion polymorphism in intron 7 (PROGINS)	nP = 72 (24 male, 48 female, 50 with agoraphobia), nC = 452 (253 women 199 men). Sweden.	Allele rs10895068:A is more frequent in patients (*p* = 0.01). Association is found in female subsample (*p* = 0.0009), male subsample lacks association. PROGINS insertion was also studied, no association found.	[57]
*RGS2* (regulator of G protein signaling 2)	RGS2 protein regulates G protein signaling activity and modulates receptor signaling neurotransmitters involved in the pathogenesis of anxiety diseases.	rs10801153	(1) nP = 239 (143 female, 96 male), nC = 239 (143 female, 96 male), Germany. (2) nP = 292 (216 female, 76 male), nC = 292 (216 female, 76 male), Germany.	Allele rs10801153:G is associated with PD in a combined and 2nd sample (*p* = 0.017 for the combined sample). 5 SNPs (rs16834831, rs16829458, rs1342809, rs1890397, rs4606) in *RGS2* were also studied, no association found.	[58]
		Dinucleotide repeats [(GT)12–18/(CT)4–5] in the 5′-regulatory region	nP = 87 (59 female, 20 male), nC = 87 (59 female, 20 male) Germany. nP = 124 (30 female, 43 male), nC = 124 (30 female, 43 male) Italy.	Association was found in German female subsample (*p* = 0.01), but not in Italian sample (*p* = 0.54).	[59]
*SLC6A4* (solute carrier family 6 member 1, SERT)	The norepinephrine transporter is responsible for the reuptake of norepinephrine and dopamine into the presynaptic nerve endings.	rs2242446, rs11076111, rs747107, rs1532701, rs933555, rs16955584, rs36021	nP = 449 (321 female, 128 male), nC = 279 (223 female, 56 male). Denmark, Germany.	7 SNPs are associated with PD with *p*-value from 0.0016 to 0.0499 in female subsample and patients with PD with agoraphobia (*n* = 226) subsample. Strongest associations are with rs2242446 (*p* = 0.0018) and rs747107 (*p* = 0.0016). Haplotype rs2242446, rs8052022 is associated with PD (*p* = 0.0022): rs2242446:T, rs8052022:T variant is protective, rs2242446:C, rs8052022:T is risk. The male subsample lacks association. A total of 29 SNPs were studied.	[60]
*TPH2* (tryptophan hydroxylase)	*TPH2* encodes key enzyme limiting the rate of transmission of nerve impulses. It is also involved in serotonin synthesis.	rs4570625	nP = 108 (58 males, 50 females), nC = 247 (125 males, 122 females). South Korea.	Allele rs4570625:T is less frequent in patients and is very frequent in female controls (*p* = 0.016).	[61]
		rs1386483	nP = 213 (163 females, 50 males), nC = 303 (212 females, 91 male). Estonia.	SNP rs1386483 is associated with PD in female with pure PD phenotype (without comorbidity) subsample (*p* = 0.01).	[18]
*TMEM132D* (transmembrane protein 132D)	Transmembrane protein 132D is associated with the severity of anxiety symptoms in psychiatric patients. A gene may be involved in anxiety (anxiety-related behavior).	rs4759997	(1) *HLA-DRB1*13:02*-positive subjects nP = 103; nC = 198. Japan. (2) *HLA-DRB1*13:02*-negative subjects nP = 438; nC = 1.341). Japan.	SNP rs4759997 is associated with PD in patients without the HLA-DRB1*13:02 allele (*p* = 5.02 × 10^−6^), but not in carriers of this allele. 9 SNPs with weaker associations were found.	[62]
		rs900256, rs879560, rs10847832	nP = 909, nC = 915 (three combined samples). Germany.	GWAS showed haplotype TA (rs7309727:T, rs11060369:A) association with PD. A replication study found several symptom severity associations, the strongest one with rs900256:C (*p* = 0.0003). Other risk alleles include rs879560:A (*p* = 0.0006) and rs10847832:A (*p* = 0.0007).	[63]
		rs10847832 and rs4680 (*COMT*), rs3219151 (*GABRA6*)	nP = 189 (105 males, 84 females)), nC = 398 (208 males, 190 females). Japan.	Allele rs10847832:A is associated with PD (*p* = 0.03). Patients carrying genotype rs11060369:AA (*TMEM132D*), and allele rs3219151:T (*GABRA6*) had significantly stronger response to frightening image demonstration (both *p* < 0.01). A total of 5 SNPs were studied. See also *COMT* and *GABRA6*.	[24]
		rs7309727, rs11060369	nP = 1670, nC = 2266: (1) nP = 102, controls nC = 511. Aarhus. nP = 141, nC = 345. Copenhagen. nP = 217. Gothenburg. Denmark and Sweden. (2) nP = 217, nC = 285. Estonia. (3) nP = 38, nC = 40. Iowa. nP = 127, nC = 123. Toronto. USA/Canada. (4) nP = 77, nC = 192. USA. (5) nP = 760, nC = 760. Japan.	Five patient samples were studied. In the combined sample: Allele rs7309727:C is found to be risk (*p* = 1.1 × 10^−8^). Haplotype rs7309727:C, rs11060369:C is found to be risk (*p* = 1.4 × 10^−8^). No significance in the Japanese sample. A total of 4 SNPs were studied.	[64]

^1^ Official gene name according to NCBI database. ^2^ Possible reasons for the association, including possible molecular/cellular level mechanisms. ^3^ Polymorphic locus identifiers (if present), if associated with a disease in conjunction with another marker, this marker is also included. ^4^ Sample parameters, including country, gender ratio, and other possibly important information (i.e., concomitant diseases) for patients and controls if this information is given in the original paper. ^5^ A summary of the authors’ findings regarding the association of polymorphisms with PD. ^6^ Link to the original paper.

**Table 2 genes-11-01310-t002:** Genes with no significant association with PD diagnosis reported.

Gene Name ^1^	Marker ^2^	Sample ^3^	Commentary ^4^	Reference ^5^
*ACE* (angiotensin I converting enzyme)	rs4646994	nP = 101 (37 males, 64 female), nC = 184 (74 males, 110 females). Japan.	No significant associations were found.	[65]
	rs4646994	nP = 123 (81 female, 42 male), nC = 168 (104 female, 64 male). Turkey.	No significant associations were found. Ins allele is more frequent in male PD subsample but does not reach the significance threshold.	[66]
*ADORA2A* (adenosine A2a receptor)	rs5751876	nP = 104 (43 male, 61 female), nC = 192 (88 male, 104 female). China.	No significant associations were found.	[67]
	rs1003774, rs743363, rs7678, rs1041749, SNP-4 C/T 78419	nP = 153 (70 probands from families with PD, 83 child–parent ‘trios’. USA.	No association with single SNPs. Haplotype (rs1003774, SNP-4 C/T 78419, rs1041749) was closest to significance.	[68]
*ASIC1* (acid-sensing ion channel subunit 1, *ACCN2*)	D17S1294–D17S1293, rs8066566, rs16589, rs16585, rs12451625, rs4289044, rs8070997, rs9915774	nP = 13, nC = 43. Faroe Islands, Denmark. nP = 243, nC = 645. Denmark.	A total of 38 SNPs were studied. All SNPs were first analyzed by GWAS in a Faroese sample. Several genotypes showed and a D17S1294–D17S1293 segment showed an association that did not survive further testing. Danish sample showed a nominally significant association with rs9915774 (*p* = 0.031) which didn’t survive Bonferroni correction.	[69]
*BDNF* (brain derived neurotrophic factor)	rs6265	nP = 109 (39 males, 70 female), nC = 178 (75 males, 103 females). Japan.	No significant associations were found.	[70]
*CAMKK2* (calcium/calmodulin-dependent protein kinase kinase 2)	rs3817190	nP = 179, nC = 462. Germany.	A total of three genes were studied (*P2RX7*, *P2RX4*, and *CAMKK2*). No significant associations were found. However, patients with rs3817190:AA in *CAMKK2* show more severe panic attacks and more distinguished agoraphobia. See also *P2RX7*. Patients with rs1718119:AA in *P2RX7* show more severe symptoms of PD with agoraphobia as well.	[71]
*CCK* (cholecystokinin)	−36C- > T in *CCK* promoter	nP = 98, nC = 247. Japan.	No significant associations were found.	[72]
*CCKBR* (cholecystokinin B receptor)	microsatellite CT repeat in the CCKBR	nP = 71 (39 males, 32 females), nC = 199 (111 males 88 females), Japan.	No significant associations were found.	[73]
*CHRNA4* (cholinergic receptor nicotinic α 4 subunit)	3 SNPs	*n* = 88 (31 males, 57 females), Germany.	No significant associations were found.	[74]
*COMT* (catechol-O-methyltransferase)	rs165599, rs737865	(1) 121 nuclear family. Canada. (2) nP = 89 (59 females, 30 males), nC = 89. Canada.	No significant associations were found.	[27]
	rs4680	nP = 26, nC = 26. South Korea.	No significant associations were found.	[75]
*DRD2* (dopamine receptor D2)	141C ins/del polymorphism, rs1799732, rs12364283, rs1800497	nP = 99 (76% females), nC = 104 (77% females). Poland.	No significant associations were found.	[76]
*ELN* (elastin)	Single-Stranded Conformational Polymorphism	23 independent probands. USA.	No significant associations were found.	[77]
*GABRA1*	Repeat polymorphisms in γ-aminobutyric acid receptor genes	1) 21 multiplex panic disorder pedigrees. cP = 252 (107 male, 145 female). The Midwestern United States. 2) 5 multiplex panic disorder pedigrees. cP = 125. Iceland.	A total of 8 genes (*GABRA1*, *GABRA2*, *GABRA3*, *GABRA4*, *GABRA5*, *GABRB1*, *GABRB3*, *GABRG2*) were studied. No significant associations were found.	[78]
*GABRA2*				
*GABRA3*				
*GABRA4*				
*GABRA5*				
*GABRB1*				
*GABRB3*				
*GABRG2*				
*HTR1A* (serotonin 1A receptor)	C(−1019)G	nP = 94 (52 male, 42 female), nC = 111 (52 male, 59 female). South Korea	No significant associations were found.	[79]
	rs6295	nP = 194, nC = 172. South Korea.	No significant associations were found.	[80]
*HTR2A* (5-hydroxytryptamine receptor 2A)	1438A/G, rs6313	nP = 107, nC = 161. South Korea.	No significant associations were found. The authors suggest further studies of 1438:G and rs6313:C (102:C) alleles which may be associated with symptom severity.	[81]
	rs6313	nP = 105 (66 female, 39 male), nC = 130 (89 female, 41 male). Turkey.	No significant associations were found.	[30]
	T102C	nP = 35, nC = 87. Germany.	No significant associations were found.	[82]
	T102C	(1) PD: nP = 94, nC = 94. Canada. (2) PD: nP = 86, nC = 86. Germany. (3) PD with agoraphobia: nP = 74, nC = 74. Canada. (4) PD with agoraphobia: nP = 59, nC = 59. Germany.	No significant associations were found.	[27]
	rs2296972	nP = 154 (68.6% females), nC = 347 (70.8% females)	No association with PD diagnosis. A number of rs2296972:T alleles are associated with the severity of PD (*p* = 0.029). A total of 15 SNPs were studied.	[83]
*IKBKE* (inhibitor of nuclear factor kappa B kinase subunit epsilon)	rs1539243, rs1953090, rs3748022, rs15672	nP = 190 (44 male, 146 female), nC = 371 (111 male, 260 female). Estonia.	No significant associations were found. Associations with rs1539243:T, rs1953090:C, and rs11117909:A alleles lose significance after correction for multiple testing. Associations with risk haplotype rs1539243:T, rs1953090:A, and haplotype rs1539243:C, rs1953090:C loses significance after multiple testing as well.	[45]
*IL10* (interleukin 10)	rs1554286	nP = 210, nC = 356. Estonia.	Allele rs1554286:C (*IL10*) is associated with a combined patient sample with MDD and PD. Association lost significance after the permutation test.	[44]
	rs1800896	nP = 135 (71 male, 64 female), nC = 135 (54 male, 81 female). South Korea.	No significant associations were found. However, rs180089:G is more frequent in the control group for females.	[84]
*LSAMP* (limbic system associated membrane protein)	rs1461131, rs4831089, rs9874470, rs16824691	nP = 196 (46 male, 150 female), nC = 364 (112 male, 252 female). Estonia.	Alleles rs1461131:A and rs4831089:A are associated with PD, but lose significance after correction for multiple testing. Haplotype rs9874470:T, rs4831089:A, rs16824691:T, rs1461131:A is protective, but loses significance after permutation tests. Authors suggest additional tests for these polymorphisms.	[85]
*MAOA* (monoamine oxidase A)	T941G	nP = 38 (PD = 38 (female = 21, male = 17), MD = 108 (female = 80, male = 28)), nC = 276 (female = 132, male = 144). Germany.	No significant associations were found. In the male subsample, 84.6% of patients carried the *MAOA* 941T allele. This result never reached the significance threshold due to the small sample size.	[86]
*P2RX7* (purinergic receptor P2X 7)	rs1718119	nP = 179, nC = 462. Germany.	A total of three genes studied (*P2RX7*, *P2RX4*, and *CAMKK2*). No significant associations found. However, patients with rs1718119:AA in *P2RX7* show more severe symptoms of PD with agoraphobia.	[71]
*RGS2* (regulator of G protein signaling 2)	rs2746071, rs2746072, rs12566194, rs4606, rs3767488	nP = 186 (94 males and 92 females), nC = 380 (164 males, 216 females). Japan.	No significant associations found. Haplotype AC: rs2746071-rs2746072 was nominally associated and lost significance after tests.	[87]
*SLC6A4* (solute carrier family 6 member 4)	5-HTTLPR S/L (short/long) 44bp insertion/deletion	nP = 244 (143 male, 101 female), nC = 227 (102 male, 125 female). South Korea.	No significant associations found.	[88]
		(1) nP = 88. Germany. (2) nP = 73. Italy. Matched control.	No significant associations found.	[89]
		nP = 95 (53 female, 42 male). Italy.	The authors tested the hypothesis that the heterogeneity of CO2 reactivity in patients with PD and its possible relation to 5-HTT promoter polymorphism. No association found.	[90]
		nP = 67 (53 female, 14 male). Brazil.	No significant associations found. Asymptomatic patients with panic disorder. No association on the MMPI scales between different genotype classifications and allele analyses.	[91]
		nP = 194, nC = 172. South Korea.	No significant associations found. However, the number of separation life events and their interaction with 5-HTTLPR showed a statistically significant effect on PD.	[80]
*TPH2* (tryptophan hydroxylase 2)	rs1800532	nP = 244 (143 male, 101 female), nC = 227 (102 male, 125 female). South Korea.	No significant associations found.	[88]
	218 A/C	nP = 107, nC = 161. South Korea.	No significant associations found.	[81]
	rs4570625 rs4565946	nP = 134, nC = 134. Germany.	No significant associations found.	[92]
*TPH* (tryptophan hydroxylase)	A218C	nP = 35, nC = 87. Germany.	No significant associations found.	[82]

^1^ Official gene name according to NCBI database. ^2^ Polymorphic locus identifiers (if present). ^3^ Sample parameters, including country, gender ratio, and other possibly important information (i.e., concomitant diseases) for patients and controls if this information is given in the original paper. ^4^ A summary of the authors’ findings regarding the association of polymorphisms with PD. ^5^ Link to the original paper.

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
