# Peer review of "Genetic Biomarkers of Panic Disorder: A Systematic Review"

_genes, 2020, doi:10.3390/genes11111310_

Round 1
Reviewer 1 Report
line 9: put "(PD)" after "panic disorder"
line 20: the same
line 27: [6] estimates heritability at 0.48
line 30: use dot, not comma before decimals; the same for the entire article
Being a review, the research must respect PRISMA guideline. Even if you used PathwayStudio’s ResNet database, every article should be revised by two authors and any differences between them must arbitrated by a third author or by discussion between the two authors.
How many articles were reviewed? How many articles were excluded? Do the PRISMA flowchart.
Author Response
Dear Reviewer 1, thank you for your careful review of our work. We revised our paper and took your remarks into account.
As to our algorithm, in no means our protocol was fully automated. Each article taken into analysis after basic exclusion using Pathway Studio tools was carefully reviewed by authors manually and discussed if any disagreements occurred. This applies not only to articles reporting associations but also to articles in “No associations” table. To avoid any misinterpretations of our work, we included detailed protocol used in current study. We also revised our manuscript and data provided in accordance to PRISMA guidelines. Number of included and excluded articles is also available now in the text and Figure composed according to PRISMA flowchart.
Reviewer 2 Report
The manuscript reviews available empirical data about biomarkers of a frequent anxiety disorders: panic disorder (PD). This review is supported in specific database, Pathway Studio. Biomarkers of the mental disorders is a relevant aim with clear implications with a better knowledge of disease, precise diagnosis and its treatment (especially with psychodrugs). In that sense, manuscript may be interesting and appropriate for Genes journal.
- Major concerns about this item is with its formal presentation and with the extent of data extracted:
- I think title may include a term (or terms) pointing out the manuscript deals with a literature review.
- Abstract may include number of research finally selected and may provide specific results about main biomarkers associated (and no associated) with PD.
- I miss a clear explanation why biomarkers for PD is a relevant information (and not for anxiety disorders, in general, or other anxiety disorders).
- A ‘method’ section is needed. Something is said in the initial paragraphs of ‘results’ section, but information is insufficient. Genes' norms propose PRISMA guidelines for systematic reviews and meta-analysis. Despite authors uses a curated database (Pathway Studio) with a specific protocol, I think manuscript can be as close as possible to the PRISMA guidelines. It implies information about precise inclusion/exclusion criteria, terms used to select investigation finally included in the review. How selection was done (process). If analysis of experimental biases for each research was done... Also, tables are correct and provide relevant information, but, again, its format can be as close as possible to PRISMA.
- Finally, and especially relevant, I think results' discussion lacks implications for PD. Identification of biomarkers have clear implications. In the case of mental disorders (and not only for future studies). Its implications can reach to the use of preventive strategies, early treatments and/or psychodrugs' efficiency. I think manuscript can be more interesting if authors comment these implications and include it in its conclusions.
Author Response
Dear Reviewer 2, thank you for your review and appreciation of our work. We revised our paper and took your remarks into account.
We included a term clearly pointing out that manuscript deals with a literature review and added required information to abstract and conclusion sections.
As to importance of PD biomarkers, anxiety disorders have various different nosologies. We primary selected this disorder due to our professional and scientific interests considering its’ differences from generalized anxiety and other anxiety disorders. However, as it is pointed out in the text, PD is one of the most widespread anxiety disorders dramatically affecting patients’ quality of life.
We fully agree with your remark regarding methods used. We re-written this section, added detailed protocol description and charts according to PRISMA guidelines. Performed manual and automated steps, number of included and excluded articles and other relevant information is available now in the Methods section and on the Figure composed according to PRISMA flowchart. We believe that this information will be useful for other researchers willing to conduct similar work in other fields or use our review as a base for their own research.
We also expanded Discussions section adding possible implications of PD biomarkers for further studies and diagnostics.
Round 2
Reviewer 1 Report
Accept in present form.